# Development of Novel Class of Phenylpyrazolo[3,4-*d*]pyrimidine-Based Analogs with Potent Anticancer Activity and Multitarget Enzyme Inhibition Supported by Docking Studies

**DOI:** 10.3390/ijms241915026

**Published:** 2023-10-09

**Authors:** Ahmed K. B. Aljohani, Waheed Ali Zaki El Zaloa, Mohamed Alswah, Mohamed A. Seleem, Mohamed M. Elsebaei, Ashraf H. Bayoumi, Ahmed M. El-Morsy, Mohammed Almaghrabi, Aeshah A. Awaji, Ali Hammad, Marwa Alsulaimany, Hany E. A. Ahmed

**Affiliations:** 1Pharmacognosy and Pharmaceutical Chemistry Department, College of Pharmacy, Taibah University, Al-Madinah Al-Munawarah 41477, Saudi Arabia; akjohani@taibahu.edu.sa (A.K.B.A.); mhmaghrabi@taibahu.edu.sa (M.A.); msulaimany@taibahu.edu.sa (M.A.); 2Pharmaceutical Organic Chemistry Department, Faculty of Pharmacy, Al-Azhar University, Cairo 11884, Egyptm.seleem@azhar.edu.eg (M.A.S.); m.elsebaei@azhar.edu.eg (M.M.E.); ashraf.bayoumi@azhar.edu.eg (A.H.B.); ahmedelmorsy232@yahoo.com (A.M.E.-M.); alihammad@azhar.edu.eg (A.H.); 3Pharmaceutical Chemistry Department, College of Pharmacy, The Islamic University, Najaf 54001, Iraq; 4Department of Biology, Faculty of Science, University College of Taymaa, University of Tabuk, Tabuk 71491, Saudi Arabia; aawaji@ut.edu.sa

**Keywords:** phenylpyrazolo[3,4-*d*]pyrimidine, tyrosine kinase, cell cycle analysis, EGFR, VGFR, apoptosis, docking

## Abstract

Phenylpyrazolo[3,4-*d*]pyrimidine is considered a milestone scaffold known to possess various biological activities such as antiparasitic, antifungal, antimicrobial, and antiproliferative activities. In addition, the urgent need for selective and potent novel anticancer agents represents a major route in the drug discovery process. Herein, new aryl analogs were synthesized and evaluated for their anticancer effects on a panel of cancer cell lines: MCF-7, HCT116, and HePG-2. Some of these compounds showed potent cytotoxicity, with variable degrees of potency and cell line selectivity in antiproliferative assays with low resistance. As the analogs carry the pyrazolopyrimidine scaffold, which looks structurally very similar to tyrosine and receptor kinase inhibitors, the potent compounds were evaluated for their inhibitory effects on three essential cancer targets: EGFR^WT^, EGFR^T790M^, VGFR2, and Top-II. The data obtained revealed that most of these compounds were potent, with variable degrees of target selectivity and dual EGFR/VGFR2 inhibitors at the IC_50_ value range, i.e., 0.3–24 µM. Among these, compound **5i** was the most potent non-selective dual EGFR/VGFR2 inhibitor, with inhibitory concentrations of 0.3 and 7.60 µM, respectively. When **5i** was tested in an MCF-7 model, it effectively inhibited tumor growth, strongly induced cancer cell apoptosis, inhibited cell migration, and suppressed cell cycle progression leading to DNA fragmentation. Molecular docking studies were performed to explore the binding mode and mechanism of such compounds on protein targets and mapped with reference ligands. The results of our studies indicate that the newly discovered phenylpyrazolo[3,4-*d*]pyrimidine-based multitarget inhibitors have significant potential for anticancer treatment.

## 1. Introduction

Cancer is a significant and prevalent health issue and is one of the leading causes of death globally [1,2]. This disease is characterized by sustained proliferative potential, growth signals, and self-sufficiency, with apoptotic and antiproliferative cues resistance [3,4,5]. There have been advancements in diagnosing cancer early and the development of new treatments; however, there is still a lack of effective therapeutics for treating advanced cancers [6]. As conventional methods for treatment of cancer, radio and chemotherapy are no longer effective due to several side effects, including the unbiased destruction of body cells at comparable rates [7,8]. Thus, multiple attempts have been made to treat advanced cancer cases utilizing targeted therapies [9,10].

For targeted anticancer therapies, the pyrazolo[3,4-*d*]pyrimidine scaffold has gained great recognition due to its diverse and versatile pharmacological potential and structural similarity to ATP cofactor [11,12,13]. This similarity urges the utilization of pyrazolo[3,4-*d*]pyrimidine as a bioisosteric replacement of adenine (9H-purin-6-amine) as it would keep the fundamental interactions at the kinase domain [14,15]. The extensive role of kinases in many reported illnesses encourages extensive work to design and deliver potent analogs of pyrazolo[3,4-*d*]pyrimidine [14,16,17,18]. Several derivatives have shown a distinct growth-inhibitory activity via CDK1, CDK2, and 5-lipoxygenase enzymes’ inhibition [19,20]. Interesting drugs that bear this scaffold are Dinaciclib, ibrutinib, and roscovitine, with potent CDK2 inhibition [14,16,17,18].

The advancement of targeted therapies that aim to hinder or obstruct crucial cellular pathways in tumor growth and metastasis has led to a greater understanding of the diversity within tumors and their ability to circumvent the blockade of signaling pathways. Consequently, certain tumors may inherently display resistance or develop resistance to therapies that target a particular pathway. To address this challenge, employing a comprehensive strategy that involves the concise inhibition of multiple signaling pathways may be fruitful. This strategy can help counteract tumor resistance by blocking potential alternative paths for tumor escape [5,21,22,23,24,25,26,27].

Vascular endothelial growth factor (VEGF) and the epidermal growth factor receptor (EGFR) are complementary pathways that play a fundamental role in tumor survival and diffusion [28,29,30]. The VEGF signaling pathway is upregulated by the expression of EGFR, which contributes to cancer resistance. VEGF and EGFR can exaggerate tumors through the exertion of both indirect and direct effects on tumor cells [28]. Targeting both pathways via mono- or multi-target therapeutics demonstrates a potential clinical benefit in many cancerous conditions. It is worth noting that therapeutics that impact VEGF-related pathways may contribute to the therapeutic targeting of the EGFR pathway [21,31,32,33,34]. Dual inhibitors of VEGF/EGFR could potentially improve antitumor efficacy and overcome resistance.

1*H*-Pyrazolo[3,4-*d*]pyrimidine scaffold has been reported to be an essential pharmacophore in many anticancer agents [16,35,36], including EGFR-TKIs. Herein, given examples include compounds **1**–**8** (Figure 1), which were tested as anticancer agents with a pyrimidine-based library of the anti-EGFR-TK mechanism and have been approved by the FDA; these include first-generation examples such as erlotinib **1** and gefitinib **2**; second-generation examples such as afatinib **3** and canertinib **4**; third-generation EGFR-TKIs such as rociletinib **5**, and avitinib **6**; and clinical-phase compounds such as sapitinib **7** and dacomitinib **8**.

The promising biological effects of 1*H*-pyrazolo[3,4-*d*]pyrimidine, in which the core scaffold was decorated with a pendant *N*-linker bonded to the aryl moiety at the 4 position, are able to consistently conserve activity against targets.

Other complexed examples of pyrazolo[3,4-*d*]pyrimidine-based EGFR-TK inhibitors (**9**–**13**) were discovered, with different potency profiles based on the ATP pharmacophore model. These compounds showed very interesting anticancer activity against specific cancer cell lines [37,38,39,40] (Figure 2).

## 2. Results and Discussion

### 2.1. Structure-Based Scaffold and Compound Design

Reports revealed that EGFR-TK is a polypeptide chain and that the ATP-binding groove of EGFR-TK is a 1186-amino-acids polypeptide chain. The ATP-binding region is composed of three areas: extracellular, intracellular, and hydrophobic regions (Figure 3a) [41,42]. SAR analysis of EGFR-TKIs revealed that they have four common pharmacophoric features. First, the core structure consists of a flat nitrogenous heterocycle. This ring occupies the adenine binding pocket and forms hydrogen bonding with Met793, Thr790, and Thr854. The next feature is represented by a terminal hydrophobic head (a plain or substituted phenyl group) that interacts with hydrophobic region I. Most of the reported inhibitors also bear a middle secondary amine group, occupying the linker region between the adenine binding region and the hydrophobic region. The final feature, represented by a hydrophobic tail, connects to the flat hetero aromatic ring system which occupies hydrophobic region II.

Based on the previous findings, we aimed to synthesize new 1*H*-pyrazolo[3,4-*d*]pyrimidine derivatives with potential anticancer activity. The target of this work was the synthesis of new derivatives carrying the same essential pharmacophoric features of the reported EGFR-TKIs as compound **1** and compound **7** (Figure 3b). We used a bioisosteric replacement strategy to fill up the binding pocket in EGFR-TKIs at four different positions (Figure 3b). In the first position, we used 1*H*-pyrazolo[3,4-*d*]pyrimidine as a flat heterocyclic system as a bioisostere of the quinazoline core on compounds **1** and **7**. The suggested 1*H*-pyrazolo[3,4-*d*]pyrimidine core can fill the bulky space of the adenine-binding region (Figure 3a) [43,44]. The nitrogen atoms at the core would engage several hydrogen bonds, which would be translated into excellent EGFR-TK potency [44,45].

The second position and third regions consisted of two terminal hydrophobic regions represented by phenyl group position 1 and alkyl or aryl side chains connected to a nitrogenous spacer. The last part was the nitrogenous linker (spacer) region with various lengths and heteroatom contents (amine, hydrazide, or thiosemicarbazide). The suggested modifications we envisioned led to study the SAR of this scaffold as an anticancer agent through the inhibition of EGFR-TK.

### 2.2. Chemistry

The route adopted for the synthesis of compounds **5a**–**l** and **6**–**9** is depicted in Figure 1 and Figure 2. Synthesis was initiated via chlorination of 6-methyl-1-phenyl-1,5-dihydro-4*H*-pyrazolo-[3,4-*d*]pyrimidin-4-one (**1**) using phosphorous oxy-chloride in the presence of trimethylamine (TMA) to give the chlorinated derivative, 4-chloro-6-methyl-1-phenyl-1*H*-pyrazolo[3,4-*d*]pyrimidine (**2**). Hydrazinolysis of the 4-chloro derivative (**2**) under reflux gave the hydrazide derivative, 4-hydrazinyl-6-methyl-1-phenyl-1*H*-pyrazolo[3,4-*d*]pyrimidine (**3**). Condensation of compound **3** with appropriate aromatic aldehydes (**4a**–**h**) or acetophenones (**4i**–**l**) in ethanol and glacial acetic acid (as a catalyst) afforded the open chain Schiff base products, 4-(2-arylidenehydrazinyl)-6-methyl-1-phenyl-1*H*-pyrazolo[3,4-*d*] pyrimidine (**5a**–**l**).

Next, we used 4-hydrazinyl-6-methyl-1-phenyl-1*H*-pyrazolo[3,4-*d*]pyrimidine (**3**) as a versatile intermediate to synthesize a variety of pyrazolo-triazolopyrimidine derivatives. Thus, refluxing of **3** with triethyl orthoformate, trifluoroacetic acid, and trichloroacetic acid, respectively, gave the following tricycle compounds; 5-methyl-7-phenyl-7*H*-pyrazolo[4,3-*e*][1,2,4]triazolo[4,3-*c*]pyrimidine (**6a**), 3-trifluoromethyl (**6b**) and 3-trichloromethyl (**6c**) derivatives, respectively, were formed. While condensation of **3** with acetyl acetone afforded the open-chain Schiff base product, 4-((6-Methyl-1-phenyl-1H-pyrazolo[3,4-*d*]pyrimidin-4-yl)imino)pentan-2-one (**6d**). Reaction of **3** with dihydrofuran-2,5-dione, furan-2,5-dione, and indoline-2,3-dione, respectively, afforded 1-((6-methyl-1-phenyl-1*H*-pyrazolo[3,4-*d*]pyrimidin-4-yl)amino)pyrrolidine-2,5-dione (**7**), 1-((6-methyl-1-phenyl-1*H*-pyrazolo[3,4-*d*]pyrimidin-4-yl)amino)-1*H*-pyrrole-2,5-dione (**8**), and 3-(2-(6-methyl-1-phenyl-1*H*-pyrazolo[3,4-*d*]pyrimidin-4-yl)hydrazono)indolin-2-one (**9**), respectively, as shown in the Figure 2.

Finally, interaction of the hydrazinyl derivative (**3**) with several isothiocyanates, namely, ethyl isothiocyanate, propyl isothiocyanate, butyl isocyanate, vinyl isothiocyanate, and phenyl isothiocyanate, in butanol (20 mL) under reflux afforded the *N*-alkyl/aryl-2-(6-methyl-1-phenyl-1*H*-pyrazolo[3,4-*d*]pyrimidin-4-yl)hydrazinecarbothioamides (**9a**–**e**), respectively, as shown in the Figure 2.

### 2.3. Anticancer Assay

The antiproliferative activity of the synthesized library was examined against three human cancer cell lines; breast (MCF-7), colon (HCT-116), and liver (HepG_2_) utilizing an MTT assay with the human diploid fibroblasts (WI-38) normal cell line as a comparison [46,47,48]. The data are summarized in Table 1. According to our design, the tested compounds are classified into four categories based on the length and the chemotype of the spacer (Figure 3). In the first category, we used monosubstituted hydrazone linker. Among the tested derivatives in this category, compound **5b** with p-hydroxyphenyl showed the most promising activity against the three cell lines when compared with the reference drug. Additionally, the O-methylated congeners, compounds **5d** and **e**, maintained promising activity against HCT-116 cells with IC_50_ 9.87 and 8.15 µM, respectively. The second category is represented by monosubstituted hydrazone linker. In this category, compound **5a**, carrying the unsubstituted phenyl group, demonstrated noticeable activities against the three cell lines.

Possessing various chemical characteristics on scaffold dramatically ameliorated cytotoxic activity. In the third category, we tethered the linker to the pyrimidine ring to provide the triazolo[4,3-*c*]pyrimidine core. Using this strategy, we aimed to reduce rotation at this part to reduce the entropic penalty of the binding of the compound to EGFR-TK. The three derivatives showed moderate cytotoxic activities. Next, we used four heterocyclic substitutions to build the fourth category of our compounds (**6d**–**7**). None of these molecules exhibited any meaningful activity against the three cell lines. In the last category, we used a thiosemicarbazide linker connected to various aliphatic and aromatic side chains (**9a**–**e**) with prominent activity for analogs **9a** and **9e**. Collectively, this series displayed the best activity against the three tested cell lines in comparison with the other categories (Table 1). In addition, compound **9a**, with ethyl side chain, overrides the activity of Doxorubicin against the HCT-116 cell line. In addition, normotoxicity was assessed for the most active analogs—**5b**, **5i**, and **9e**—on the WI38 cell line, and they exhibited a very low effect, with IC_50_ greater than 35 µM, which indicated a good safety profile.

Next, we investigated the cytotoxic activity of selected derivatives against the same cell lines using the extended treatment strategy. Using this strategy, we aimed to examine the resistance ability of the tested cell lines to our compounds after incubation for a long time. We realized that in comparison with doxorubicin, some tested compounds exhibited a time-dependent cytotoxic activity; for example, compound **5i**. On the other hand, we noted a transient partial loss of cytotoxic activity 48 h post-treatment, followed by improved activity in the last time point. Table 2 and Figure 4 summarize the relative cytotoxic activity of the selected derivatives.

### 2.4. In Vitro Cancer Related Targets Inhibition Analysis

Multi-targeting is a useful anticancer strategy with superior therapeutic attributes. A variety of tumors, such as aggressive breast cancers, offer the overexpression of different cellular enzyme targets which are responsible for large tumor size, poor differentiation, and poor clinical outcomes [49,50,51,52]. Triazole hybrids were discovered to be multi-target EGFR^WT^-, EGFR^T790M^-, VEGFR-2-, and Topo II based-inhibitors, and they were evaluated for anticancer activity [53,54]. Moreover, examples of Pyrazolo[3,4-*d*]pyrimidine-based multitarget anticancer inhibitors were reported [55]. As seen in the data for the cytotoxicity of the newly synthesized analogs, only active candidate compounds **5b**, **5i**, and **9e** were tested against a panel of cancer-related targets, namely, EGFR, VEGFR-2, and Topo-II, compared to reference drugs. The results are summarized in (Table 3). We found that compound **5b** strongly and selectively inhibited EGFR activity in the lower range, with 20-fold target selectivity indices over VGFR-2 and 108-fold over Topo-II. Moreover, compound **5i** showed low selective and potent inhibition of EGFR^WT^ and EGFR^T790M^ over VGFR-2 and Topo-II, with 20/66 selectivity indices. Compound **9e** was moderately active against VGFR-2 and Topo-II targets, with selectivity indices of 21/31 over EGFR^WT^. Overall, the scaffold pyrazolo[3,4-*d*]pyrimidine analogs showed different degrees of multitarget potent inhibitory activity, which might contribute to the discovery of anticancer agents.

### 2.5. Cell Cycle Analysis

Based on its antiproliferative activity, compound **5i** was designated for further studies to explore its effect on the induction of apoptosis in the A549 cell line [56]. We used this assay to elucidate the relationship between the proliferation inhibition and the cell cycle arrest, s well as to determine the biological phase in which the molecule interferes with cell growth. The cells were treated with 1 μM of **5i**, and we used DMSO as a negative control. As presented in Figure 5, the data showed obvious interference with the native cell cycle distribution. Exposure of MCF-7 cells to our compound caused a decrease in the proportion of cells in the G0/G1 phase (from 57.39% to 49.63% when compared with the control). Moreover, it showed a slight increase in cell percentage in the S phase (from 33.97% to 43.12%), accompanied by a slight increase in the percentage of cells at the G2/M phase of the cell cycle (from 8.64% in the control to 7.25%) and a significant increase in the pre-G1 phase of the cell cycle (from 1.79% in the control to 36.06%). Collectively, these results indicated that compound **5i** can lead to apoptosis through arresting the G1/S phase of cell cycle.

### 2.6. Annexin V-FITC Apoptosis Assay

We used annexin V-binding studies via flow cytometer to confirm the apoptosis induction by our compound. The apoptotic nature of **5i** against MCF-7 cells was tested via flow cytometry detection after double-staining with Annexin V-FITC and propidium iodide (PI) [57]. The results demonstrated that the treatment of A549 cells with our compound for 48 h increased the early apoptosis ratio (lower right quadrant of the cytogram) from 0.44% to 12.62% and increased the late apoptosis ratio (higher right quadrant of the cytogram) from 0.17% to 19.11%, indicating that **5i** can induce MCF-7 cells apoptosis. In addition, treatment of MCF-7 cells with our compound for 24 h resulted in 36.06% of apoptotic cells (early + late) versus 1.79% of apoptotic cells in the untreated control (Figure 6). These results demonstrated that **5i** might inhibit cell growth through cell apoptosis induction.

### 2.7. EGFR/VGFR2 Target Docking Simulations

The integration of experimental and computational methods is an attractive strategy by which to design and optimize successful drug candidates [32,34,51,52]. Based on the biological activity of **5i**, we carried out molecular docking studies as a crucial step to understand the mode of interaction of our selected molecule. First, a docking study was performed against EGFR (PDB ID: 1M17) (Figure 7a). Our compound was compared with erlotinib (compound **1**, Figure 1), the known EGFR inhibitor. Previous studies [34,58] have indicated that erlotinib causes EGFR inhibition by binding to the site occupied by ATP during phosphotransfer. The N^1^ of the quinazoline accepts the hydrogen bond from the Met769 amide nitrogen. It has been reported that 1-diphenyl-4,5-dihydro-1*H*-pyrazolo[3,4-*d*]pyrimidin-6-amine scaffold has successfully tolerated the EGFR pocket [59]. Our data indicated that compound **5i** conserved the same interaction with Met769 as the reference molecule. The N^1^ of the pyrazole ring formed a hydrogen bond with Met 769, with a distance of 0.8 Å. Furthermore, the pyrazolo[3,4-*d*]pyrimidine moiety was incorporated into pi–pi interaction with Gly772 and Cys773, respectively. The peripheral phenyl group of **5i** was buried inside hydrophobic region I. Additionally, a computational docking study was performed against VEGFR-2 (PDB ID: 3EWH) in comparison with a pyridyl-pyrimidine benzimidazole inhibitor (Figure 7b). The binding pattern of our molecule includes the pi–pi interaction of the core phenyl group with Leu1035 and the pyrimidine moiety with Lys868. Moreover, the hydrazide linker showed hydrogen bonding with Glu885, with a distance of 2.86 Å. This interaction imitates the same binding of the reference molecule with Glu885. A comparison of our molecule with both reference molecules through 2D molecular alignment revealed that **5i** occupies the same space in both receptors. In addition, a mapping experiment on an active **5i** analog referencing selective EGFR and VGFR-2 compounds led to the discovery of conserved molecular motifs that contributed strongly to the binding mechanism and hence to biological activity (Figure 8).

### 2.8. Prediction of Drug-Likeness and ADME Properties

We used the OSIRIS Property Explorer to predict any potential side effects of our molecules, such as mutagenic, tumorigenic, irritant, and reproductive effects. Moreover, we tested a group of drug-relevant properties, including cLogP, LogS (solubility), MW, drug-likeness, and overall drug-score. The in silico physicochemical and toxicological analyses were carried out with the OSIRIS Property Explorer program with respect to the two potent active analogs, **5b** and **5i,** compared to the two reported anticancer drugs. Properties with a high risk of undesirable effects, such as mutagenicity, tumorigenicity, irritant effects, and effects on reproductive physiology, as well as drug-relevant properties, including cLogP, LogS (solubility), MW, drug-likeness, and overall drug-score, are scored and color-coded, as shown in Table 4. The data of toxicity risks (shown in colors as red (high risk) and green (zero risk)) indicate a behavior consistent with the compound. Interestingly, the potential drug-likeness values of compounds **5i** and **5b** (4.26 and 3.97) were significantly higher than the two reference drugs, which showed negative values of −4.2 and −6.73. However, the in silico prediction of the OSIRIS Property Explorer showed that the introduction of 4-OH on the aryl ring can retain the lack of tumorigenic and mutagenic toxicity risk and enhance the druggability of compound **5i** compared to the low-active one. In addition, we can see that both **5i** and **5b** compounds present a very excellent safety profile against all side effects. Generally, the drug-score values of compounds **5b** and **5i** (0.66 and 0.58) were better than those of sorafenib and erlotinib (0.20 and 0.38). It was noted via %ABS that analog **5i** was 85% better than the others. The reasons for this good analysis are the absence of reactive functional groups within the chemical structure, the chemical stability, and the simplicity in the formula.

## 3. Conclusions

In this work, we applied a well-established chemical approach to the design and synthesis of phenylpyrazolopyrimidine-based analogs. In the protocol, we synthesized four different series of hydrazide, methylhydrazide, closed ring systems, and thiourea derivatives of high accessibility and good yields. The resulting compounds were evaluated for cytotoxic activity using an MTT assay against three malignant cell lines. Moreover, exemplary compounds were selected for mechanistic analysis using EGFR, VGFR-2, and Top-II enzymatic assay and cell cycle and apoptotic analyses. Most of the synthesized derivatives exhibited greater potency and selectivity performance than the reference inhibitor. Compounds **5b**, **5i**, and **9e** were the most potent analogs, with IC_50_ values of 3–10 µM cytotoxicity. Moreover, a detailed mechanistic analysis on the cancer cell line revealed that when compared to the positive control medication, most of the compounds had stronger antiproliferative activity. In MCF-7 cancer cells, compound **5i** also enhanced apoptosis, produced cell cycle arrest at the G2/M phase, and caused DNA fragmentation. Molecule **5i** appears to offer a lot of potential as a novel multi pyrazolopyrimidine-based lead compound for the identification of new anticancer medicines that target EGFR/VGFR-2 enzymes, based on our findings. In future studies, we intend to keep optimizing this molecule to create chemical entities with great anticancer activity and better selectivity to be advanced to preclinical studies. We will also use this molecule as a starting point from which to develop a combination therapy.

## 4. Experimental Section

### 4.1. Chemistry

#### 4.1.1. General Procedures

All solvents and reagents used in this work were utilized as received from suppliers unless otherwise noted. Melting points were determined on a Stuart^TM^ digital melting point apparatus (Stone, UK) and are uncorrected. The IR spectra were recorded on a Jasco FT/IR 460 plus spectrophotometer using KBr discs. The ^1^H-NMR (400 MHz) and ^13^C-NMR (100 MHz) spectra were recorded on a Varian Mercury VXR-400 NMR 300 MHz using DMSO-*d*_6_ as a solvent, and mass spectra were measured using a Hewlett Packard 5988 spectrometer. Elemental analysis was conducted at the Regional Centre for Mycology and Biotechnology (RCMP), Al-Azhar University, Cairo, Egypt. Reaction progress and purity of the synthesized molecules were monitored via thin-layer chromatography (TLC), utilizing Merck precoated silica gel 60 F_254_ aluminum sheets. The reported yields refer to isolated compounds after purification. All spectral data are present in Appendix A.

##### 6-Methyl-1-phenyl-1,5-dihydro-4H-pyrazolo[3,4-*d*]pyrimidin-4-one (**1**)

Compound 1 was prepared according to the procedure in [60].

##### 4-Chloro-6-methyl-1-phenyl-1H-pyrazolo[3,4-*d*]pyrimidine (**2**)

Compound 6 was prepared according to the procedure in [61].

##### 4-Hydrazinyl-6-methyl-1-phenyl-1H-pyrazolo[3,4-*d*]pyrimidine (**3**)

A mixture of 4-chloro-6-methyl-1-phenyl-1*H*-pyrazolo[3,4-*d*]pyrimidine (**2**) (0.01 mol) and hydrazine hydrate (99%, 5 mL, 0. 1 mol) was refluxed for 8 h. After cooling, the formed precipitate was filtered, washed with hot ethanol (95%, 3 mL), and crystallized from isopropanol to yield compound **3**: yellow solid; yield 73%; m.p. 236–238 °C; IR (KBr) *υ* (cm^−1^): 3444–3352 (NH_2_), 3190 (NH), 1660 (C=N), 1560 (C=C); ^1^H NMR (DMSO-*d*_6_) δ ppm: 8.26,(d, 2H, *J* = 7.90 Hz, phenyl-H2, H6), 8.17 (s, 1H-Ar-H C3-H-pyrazol), 7.56–7.60 (m, 3H, phenyl-H3, H4, H5), 7.54 (s, 1H, NH, exchanged with D_2_O), 4.73 (brs, 2H, NH2, exchanged with D_2_O), 2.42 (s, 3H, pyrimidine-CH3); ^13^CNMR (DMSO-*d*_6_) δ (ppm): 164.5, 158.06, 156.04, 148.00, 134.03, 129.03, 127.13, 126.02, 119.90, 103.30, 26.20, 24.50.; MS (*m*/*z*): 240.27 (M+, 8.95%), 171 (100%); Anal. Calc. for C_12_H_12_N_6_ (242.27): C, 59.59; H, 5.03; N, 34.98%. Found: C, 60.11; H, 5.15; N, 34.99%.

##### General Procedure for the Synthesis of 4-(2-Arylidenehydrazinyl)-6-methyl-1-phenyl-1H-pyrazolo[3,4-*d*]pyrimidine (**5a**–**l**)

A mixture of hydrazinyl derivative (**3**) (0.01 mol), appropriate aromatic aldehydes (**4a**–**h**) (0.01 mol), appropriate aromatic acetophenones (**4i**–**l**) (0.01 mol), and a catalytic amount of glacial acetic acid (0.5 mL) was heated under reflux in absolute ethanol (20 mL) for 4 h. The precipitate that formed was filtered and crystallized from ethanol to give the title compounds. The physical and spectral data of compounds **5a**–**l** were as follows:4-(2-Benzylidenehydrazinyl)-6-methyl-1-phenyl-1H-pyrazolo[3,4-*d*]pyrimidine (**5a**)

White solid; yield 70%; m.p. 160-162 °C; IR (KBr) *υ* (cm^−1^): 3377 (NH), 3058, 2782 (CH), 1600 (C=N), 1540 (C=C); ^1^H NMR (DMSO-*d*_6_) δ ppm: 8.61(s, 1H, NH, exchanged with D_2_O), 8.33 (s, 1H, -CH=N), 8.23,(d, 2H, *J* = 7.91 Hz, phenyl-H2, H6), 8.21 (s, 1H-Ar-H C3-H-pyrazol), 8.07 (d, 2H, *J* = 7.91 Hz, phenyl-CH-H2, H), 7.85–7.61 (m, 3H, phenyl-CH-H3, H4, H5), 7.59–7.51 (m, 3H, phenyl-H3, H4, H5), 2.51 (s, 3H, pyrimidine-CH_3_). ^13^C NMR (DMSO-*d*_6_) δ (ppm): 164.81, 155.96, 146.93, 139.24, 137.40, 134.46, 130.54, 129.64, 127.52, 126.79, 122.53, 121.48, 99.37, 40.60, 39.35, 25.83; MS *m*/*z* (%): 328 (M^+^, 10.26); Anal. Calcd for C_19_H_16_N_6_ (328.37): C, 69.50; H, 4.91; N, 25.59%. Found: C, 69.23 H, 4.95; N, 25.48%.

4-((2-(6-Methyl-1-phenyl-1H-pyrazolo[3,4-*d*]pyrimidin-4-yl)hydrazineylidene)methyl)phenol (**5b**)

White solid; yield 75%; m.p. 162–164 °C; IR (KBr) *υ* (cm^−1^): 3505 (br, OH), 3294 (NH), 3150 (CH aromatic), 1632 (C=N); ^1^H NMR (DMSO-*d*_6_) δ ppm: 10.79 (s, 1H, NH, exchanged with D_2_O), 9.89 (s, 1H, OH, exchanged with D_2_O), 8.60 (s, 1H, -CH=N), 8.20, (d, 2H, *J* = 7.91 Hz, phenyl-H2, H6), 8.18 (s, 1H-Ar-H C3-H-pyrazol), 8.05 (d, 2H, *J* = 7.91 Hz, phenyl-CH-H2, H), 7.89–7.68 (m, 2H, phenyl-H4, H5), 7.68–7.63 (m, 3H, phenyl-H3, H4, H5), 2.51 (s, 3H, pyrimidine-CH_3_). ^13^C NMR (DMSO-*d*_6_) *δ:* (ppm): 163.25, 161.53, 160.28, 158.39, 154.05, 151.01, 146.64, 137.81, 136.83, 131.24, 129.86, 127.11, 125.09, 122.56, 116.29, 98.53, 39.23, 23.06; MS *m*/*z* (%): 344 (M^+^, 11.55); Anal. Calcd for C_19_H_16_N_6_O (344.37): C, 66.27; H, 4.68; N, 24.40%. Found: C, 65.98; H, 4.84; N, 24.61%; purity via HPLC = 98.23%.

4-(2-(4-Chlorobenzylidene)hydrazinyl)-6-methyl-1-phenyl-1H-pyrazolo[3,4-*d*]pyrimidine (**5c**)

White solid; yield 79%; m.p. 233–235 °C; IR (KBr) *υ* (cm^−1^): 3450 (NH), 3002, 2917 (CH), 1580 (C=N); ^1^H NMR (DMSO-*d*_6_) δ ppm: 11.7 (s, 1H, NH, exchanged with D_2_O), 8.58 (s, 1H, -CH=N), 8.25 (d, 2H, *J* = 7.91 Hz, phenyl-H2, H6), 8.24 (s, 1H, Ar-H C3-H-pyrazol), 8.23 (d, 2H, *J* = 7.91 Hz, phenyl-CH-H2, H), 7.86-7.84 (m, 3H, phenyl-CH-H3, H4, H5), 7.59–7.54 (m, 2H, phenyl-H4, H5), 2.51 (s, 3H, pyrimidine-CH_3_). ^13^CNMR (DMSO-*d*_6_); δ (ppm): 167.90, 166.02, 160.28, 158.39, 154.05, 151.01, 146.64, 137.81, 136.83, 130.06, 129.86, 127.11, 125.09, 122.56, 119.29, 98.53, 39.23, 24.04; MS (*m*/*z*): 364 (M^+^+2, 10.3), 362 (M^+^, 32.0); Anal. Calcd for C_19_H_15_ClN_6_ (362.82): C, 62.90; H, 4.17; N, 23.16%. Found: C, 62.71; H, 4.19; N, 23.22%.

(4-(2-(4-Methoxybenzylidene)hydrazineyl)-6-methyl-1-phenyl-1H-pyrazolo[3,4-*d*]pyrimidine (**5d**)

Green solid; yield 90%; m.p. 227–229 °C; IR (KBr) *υ* (cm^−1^): 3418 (NH), 3054 (Ar-H), 2932 (aliph-CH), 1542 (C=N); ^1^H NMR (DMSO-*d*_6_) δ ppm: 8.94 (s, 1H, pyrimidine-NH), 8.74 (s, 1H, phenyl-CH=N, H1), 8.59 (d, 2H, *J* = 7.3 Hz, phenyl-H2, H6), 8.26 (s, 1H-pyrazol), 8.22 (d, 2H, *J* = 7.3 Hz, phenyl-C=N,H2, H6), 8.05–7.80 (m, 3H, phenyl-H3, H4, H5), 7.78 (d, 2H, *J* = 7.3 Hz, phenyl-H3, H5), 3.84 (s, 3H, phenyl-H5), 2.51 (s, 3H, pyrimidine- CH_3_); ^13^C NMR (DMSO-*d*_6_): 1641.41, 155.27, 154.57, 147.51, 139.04, 137.40, 131.04, 129.265, 129.90, 126.88, 122.56, 121.62, 99.49, 55.79, 25.50, 23.11; MS *m*/*z* (%): 358 (M^+^, 5.23); Anal. Calc. for C_20_H_18_N_6_O (358.40): C, 67.02; H, 5.06; N, 23.45%. Found: C, 67.28; H, 5.08; N, 23.48%.

4-(2-(2-Methoxybenzylidene)hydrazineyl)-6-methyl-1-phenyl-1H-pyrazolo[3,4-*d*]pyrimidine (**5e**)

Brownish solid; yield 86%; m.p. 220–222 °C; IR (KBr) *υ* (cm^−1^): 3412 (NH), 3076, 2955 (CH), 1550 (C=N); ^1^H NMR (DMSO-*d*_6_) δ ppm: 8.94 (s, 1H, pyrimidine-NH, H4), 8.74 (s, 1H, phenyl-CH=N, H1), 8.59 (d, 2H, *J* = 7.3 Hz, phenyl-H2, H6), 8.26 (s, 1H-pyrazol), 8.22(d, 2H, *J* = 7.3 Hz, phenyl-C=N, H2, H5), 8.05–7.80 (m, 3H, phenyl-H3, H4, H5), 7.78 (d, 2H, *J* = 7.3 Hz, phenyl-H3, H4), 3.84 (s, 3H, -OCH_3_), 2.51 (s, 3H, pyrimidine- CH_3_). ^13^C-NMR (DMSO-*d*_6_): 163.09, 158.17, 154.39, 148.66, 146.66, 146.94, 143.71, 138.95, 137.47, 132.29, 129.90, 122.47, 121.61, 99.32, 25.27, 18.96.; MS (*m*/*z*): 358 (M^+^, 21.58); Anal. Calcd. for C_20_H_18_N_6_O (358.40): C, 67.02; H, 5.06; N, 23.45%. Found: C, 67.28; H, 5.08; N, 23.48%.

6-Methyl-4-(2-(4-nitrobenzylidene)hydrazineyl)-1-phenyl-1H-pyrazolo[3,4-*d*]pyrimidine (**5f**)

Orange solid; yield 88%; m.p. 250–252 °C; IR (KBr) *υ* (cm^−1^): 3305 (NH), 3057, 2975 (CH), 1510 (C=N); ^1^H NMR (DMSO-*d*_6_) δ ppm: 12.52 (s, 1H, pyrimidine-NH, H4), 8.64 (s, 1H,phenyl-CH=N, H1)), 8.39 (d, 2H, *J* = 7.3, phenyl-CH=N, H2, H6), 8.37 (d, 2H, *J* = 7.3 Hz, phenyl-H2, H6), 8.23 (s, 1H-pyrazol), 8.18 (s, 2H, phenyl-H3, H5) 7.61–7.59 (m, 3H, phenyl-H3, H4, H4), 2.50 (s, 3H, pyrimidine- CH_3_). ^13^C-NMR (DMSO-*d*_6_): 147.86, 139.25, 125.99, 129.55, 128.14, 126.65, 124.54, 121.30, 40.45, 40.24, 40.03, 39.83, 39.62, 39.41, 39.20.; MS (*m*/*z*): 373 (M^+^, 24.32); Anal. Calcd for C_19_H_15_N_7_O_2_ (373.37): C, 61.12; H, 4.05; N, 26.26%. Found: C, 60.89; H, 4.06; N, 26.32%.

4-(2-(2,6-Dichlorobenzylidene)hydrazineyl)-6-methyl-1-phenyl-1H-pyrazolo[3,4-*d*]pyrimidine (**5g**)

Red solid; yield 83%; m.p. 255–257 °C; IR (KBr) *υ* (cm^−1^): 3383 (NH), 3109, 2910 (CH), 1570 (C=N); ^1^H NMR (DMSO-*d*_6_) δ ppm: 8.55 (s, 1H, pyrimidine-NH), 8.39 (s, 1H, -CH=N, H1), 8.24 (d, 2H, *J* = 7.3 Hz, phenyl-H2, H6), 8.18 (s, 1H, -pyrazol), 7.65-7.60 (m, 3H, phenyl-H3, H4, H5), 7.39 (d, 2H, *J* = 7.3 Hz, phenyl-H3, H5), 7.38 (m, 1H, phenyl-H4), 2.33 (s, 3H, pyrimidine- CH_3_). ^13^C-NMR (DMSO-*d*_6_): 165.9, 156.1, 154.8, 141.4, 139.1, 136.6, 134.1, 131.1, 131.6, 130.2, 129.9, 126.8, 99.4, 25.9; MS (*m*/*z)*: 400 (M^+^+4, 7.5), 398 (M^+^+2, 50.6), 396 (M^+^, 76.45); Anal. Calcd for C_19_H_14_Cl_2_N_6_ (397.26): C, 57.44; H, 3.55; N, 21.15%. Found: C, 57.42; H, 3.51; N, 21.21%.

6-Methyl-1-phenyl-4-(2-(-3-phenylallylidene)hydrazineyl)-1H-pyrazolo[3,4-*d*]pyrimidine (**5h**)

Orange solid; yield 95%; m.p. 241–243 °C; IR (KBr) *υ* (cm^−1^): 3317 (NH), 3112, 2886 (CH), 1550 (C=C), 1670 (C=N); ^1^H NMR (DMSO-*d*_6_) δ ppm: 10.46 (s, 1H, NH), 8.39 (d, 2H, *J* = 7.4 Hz, phenyl-H2, H6), 8.18 (s, 1H-pyrazol), 7.94 (s, 1H, N=CH), 7.60–7.52 (m, 3H, phenyl-H3, H4, H5), 7.54 (d, 2H, *J* = 7.4 Hz, phenyl-H2, H6), 7.45-7.33 (m, 3H, phenyl-H3, H4, H5), 7.22 (d, 1H, *J* =7.5 Hz, CH=CH-ph), 6.89 (d, 1H, *J* = 12 Hz, N=CH-CH=CH), 2.35 (s, 3H, pyrimidine-CH_3_). ^13^C-NMR (DMSO-*d*_6_): 164.2, 155.3, 154.5, 149.3, 139.2, 139.1, 137.4, 136.4, 129.6, 127.6, 126.9, 125.1, 122.5, 99.5, 56.5, 25.5, 18.9.; MS (*m*/*z*): 354 (M^+^, 100); Anal. Calcd for C_21_H_18_N_6_ (354.41): C, 71.17; H, 5.12; N, 23.71%. Found: C, 71.14; H, 5.18; N, 23.74%.

6-Methyl-1-phenyl-4-(2-(1-phenylethylidene)hydrazinyl)-1H-pyrazolo[3,4-*d*]pyrimidine (**5i**)

White solid; yield 90%; m.p. 245–247 °C; IR (KBr) *υ* (cm^−1^): 3140 (NH), 3058, 2900 (CH), 1610 (C=N); ^1^H NMR (DMSO-*d*_6_) δ ppm: 11.52 (s, 1H, -NH), 8.52 (d, 2H, *J* = 7.3 Hz, phenyl-H2, H6), 8.15 (s, 1H-pyrazol), 8.14 (d, 2H, *J* = 7.3 Hz, phenyl-C=N,H2, H6), 7.92-7.90 (m, 3H, phenyl-H3, H4, H5), 7.55–7.51 (m, 3H, phenyl-**C**=N, H3, H4, H5), 2.60 (s, 3H, CH_3_-C=N-), 2.50 (s, 3H, pyrimidine- CH_3_); ^13^C-NMR (DMSO, *d*_6_): 160.06, 155.46, 153.52,01, 138.53, 137.95, 136.72, 133.65, 130.35, 129.69, 128.58, 126.95, 121.89, 113.38, 99.41, 12.27, 15.19; MS (*m*/*z*): 342 (M^+^, 13.09); Anal. Calcd for C_20_H_18_N_6_ (342.40): C, 70.16; H, 5.30; N, 24.54. Found: C, 69.83; H, 5.32; N, 24.58%, purity by HPLC = 98.20%.

4-(1-(2-(6-Methyl-1-phenyl-1H-pyrazolo[3,4-*d*]pyrimidin-4-yl)hydrazineylidene)ethyl)aniline (**5j**)

White solid; yield 86%; m.p. 221–223 °C; IR (KBr) *υ* (cm^−1^): 3336, 3249 (NH_2_), 3197 (NH), 3073, 2931 (CH), 1615 (C=N); ^1^H NMR (DMSO-*d*_6_) δ ppm: 10.13 (s, 1H, -NH, H4), 8.67 (d, 2H, *J* = 7.3 Hz, phenyl-H2, H6), 8.17 (s, 1H-pyrazol), 8.15 (d, 2H, *J* = 7.3 Hz, phenyl-C=N,H2, H6), 7.86-7.84 (m, 3H, -H3, H4, H5), 7.73–7.70 (m, 2H, phenyl-**C**=N, H3, H5), 3.36 (brs, 2H, -NH_2_), 2.95 (s, 3H, CH_3_-C=N-), 2.51 (s, 3H, pyrimidine- CH_3_); ^13^C-NMR (DMSO, *d*_6_): 167.9, 166.2, 164.13, 150.7, 149.87, 148.00, 140.22, 138.74, 133.49, 129.81, 127.76, 122.35,102.83, 20.81, 17.00 14.73; MS (*m*/*z*): 357 (M^+^, 13.09); Anal. Calcd for C_20_H_19_N_7_ (357.41): C, 67.21; H, 5.36; N, 27.43%. Found: C, 67.45; H, 5.38; N, 27.48%.

4-(2-(1-(4-Methoxyphenyl)ethylidene)hydrazinyl)-6-methyl-1-phenyl-1H-pyrazolo[3,4-*d*]pyrimidine (**5k**)

Yellowish solid; yield 85%; m.p. 235–237 °C; IR (KBr) *υ* (cm^−1^): 3190 (NH), 3085, 2965 (CH), 1580 (C=N); ^1^H NMR (DMSO-*d*_6_) δ ppm: 10.12 (s, 1H, pyrimidine-**NH**, H4), 8.59 (d, 2H, *J* = 7.3 Hz, phenyl-H2, H6), 8.26 (s, 1H-pyrazol), 8.22 (d, 2H, *J* = 7.3 Hz, phenyl-C=N,H2, H6), 8.05–7.80 (m, 3H, phenyl-H3, H4, H5), 7.78 (d, 2H, *J* = 7.3 Hz, phenylC=N-, H3, H5), 3.84 (s, 3H, -OCH_3_), 2.95 (s, 3H, CH_3_-C=N-), 2.51 (s, 3H, pyrimidine-CH_3_). ^13^C-NMR (DMSO, *d*_6_): 167.9, 166.2, 155.59, 153.99, 151.89, 149.63, 141.78, 138.50, 135.09, 133.67, 129.72, 128.83, 125.20, 121.03, 105.67, 24.78, 17.47; MS (*m*/*z*): 372 (M^+^, 22.7); Anal. Calcd for C_21_H_20_N_6_O (372.42): C, 67.73; H, 5.41; N, 22.57%. Found: C, 67.79; H, 5.43; N, 22.62%.

4-(2-(1-(2-Bromo-4-chlorophenyl)ethylidene)hydrazineyl)-6-methyl-1-phenyl-1H-pyrazolo[3,4-*d*] pyrimidine (**5l**)

Orange crystals; yield 75%; m.p. 257–259 °C; IR (KBr) *υ* (cm^−1^): 3123 (NH), 3080, 2924 (CH), 1535 (C=N); ^1^H NMR (DMSO-*d*_6_) δ ppm: 10.12 (s, 1H, pyrimidine-**NH**, H4), 8.59 (d, 2H, *J* = 7.3 Hz, phenyl-H2, H6), 8.26 (s, 1H-pyrazol), 8.22 (d, 2H, *J* = 7.3 Hz, -C=N,H2, H6), 7.80–8.05 (m, 2H, phenyl_-_H4, H5), 7.78 (d, 2H, *J* = 7.3 Hz, phenyl-C=N-, H3, H5), 2.95 (s, 3H, CH_3_-C=N-), 2.51 (s, 3H, -CH_3_); ^13^C-NMR (DMSO, *d*_6_): 158.14, 153.98, 151.88, 148.93, 137.50, 134.09, 132.67, 129.72, 127.81, 127.04, 124.67, 124.07, 122.44, 121.03, 115.77, 106.66, 17.78; MS (*m*/*z*): 418 (M^+^, 37.01); Anal. Calcd for C_20_H_16_BrClN_6_ (455.74): C, 52.71; H, 3.54; N, 18.44%. Found: C, 52.97; H, 3.58; N, 18.52%.

##### General Procedure for the Synthesis of Pyrazolopyrimidine Derivatives (**6a**–**d**)

A mixture of hydrazinyl derivative (**3**) (0.01 mol) and carboxylic acid derivatives, namely, triethyl orthoformate, trifluoroacetic acid, and trichloroacetic acid or acetyl acetone/glacial acetic acid (0.01 mol/20 mL), was refluxed for 3–5 h. The reaction mixture was allowed to cool, leading to separation of the product, and then the crude product was filtered, dried, and crystallized from 1,4-dioxane to give 6a–c. The physical and spectral data of compounds **6a**–**d** were as follows:5-Methyl-7-phenyl-7H-pyrazolo[4,3-e][1,2,4]triazolo[4,3-c]pyrimidine (**6a**)

White solid; yield 74%; m.p. 250–252 °C; IR (KBr) *υ* (cm^−1^): 3040, 2967 (CH), 1595 (C=C), 1648 (C=N); 1H -NMR (DMSO-*d*_6_) δ ppm: 8.80 (s, 1H, Triazol-H2), 8.74 (d, 2H, J = 7.90 Hz, phenyl-H2, H6), 8.19 (s,1H, -pyrazol), 7.65–7.61 (m, 3H, phenyl-H3, H4, H5), 2.51 (s, 3H, pyrimedine-CH3); ^13^C-NMR (DMSO-*d*_6_) δ ppm: 159.20, 144.81, 139.09, 138.74, 134.03, 131.93, 128.13, 123.7, 120.75, 117.71, 42.77; MS (*m*/*z*): 250.27 (C_13_H_10_N_6_, 78.57%, M+); Anal. Calc. for: (C_13_H_10_N_6_),C, 62.39; H, 4.03; N, 33.05%; Found: C, 62.15; H, 4.05; N, 33.69%.

5-Methyl-7-phenyl-3-(trifluoromethyl)-7H-pyrazolo[4,3-e][1,2,4]triazolo[4,3-c]pyrimidine (**6b**)

White crystals; yield 69%; m.p. 267–269 °C; IR (KBr) *υ* (cm^−1^): 3027, 2933 (CH), 1596 (C=C), 1662 (C=N); 1H-NMR (DMSO-*d*_6_) δ ppm: 8.61 (d, 2H, *J* = 7.90 Hz, phenyl-H2, H6), 8.13 (s,1H,H-pyrazol), 7.41–7.61 (m, 3H, phenyl-H3, H4, H5), 2.54 (s, 3H, pyrimedine-CH3); ^13^C-NMR (DMSO-*d*_6_) δ ppm: 159.31, 144.81, 139.09, 138.74, 134.03, 131.93, 128.13, 123.96, 120.75, 120.39, 117.71, 42.77; MS (*m*/*z*): 318.26 (78.57%, M+); Anal. Calc. for: (C_14_H_9_F_3_N_6_): C, 52.83; H, 2.85; N, 26.41%. Found: C, 53.03; H, 2.87; N, 26.32%.

5-Methyl-7-phenyl-3-(trichloromethyl)-7H-pyrazolo[4,3-e][1,2,4]triazolo[4,3-c]pyrimidine (**6c**)

Yellow solid; yield 74%; m.p. 250–252 °C; IR (KBr) *υ* (cm^−1^): 3015, 2908 (CH), 1597 (C=C), 1685 (C=N); 1H-NMR (DMSO-*d*_6_) δ ppm: 8.67 (d, 2H, *J* = 7.90 Hz, phenyl-H2, H6), 8.16 (s,1H, -pyrazol), 7.65-7.61 (m, 3H, phenyl-H3, H4, H5), 2.51 (s, 3H, pyrimedine-CH3); ^13^C-NMR (DMSO-*d*_6_) δ ppm: 159.31, 144.81, 139.09, 138.74, 134.03, 131.93, 128.13, 123.96, 123.7, 120.75, 120.39, 117.71, 42.77; MS (*m*/*z*): 367.62 (78.57%, M+); Anal. Calc. for: (C_14_H_9_C_l3_N_6_): C, 45.74; H, 2.47; N, 22.86%; Found: C, 45.89; H, 2.46; N, 22.95%.

4-((6-Methyl-1-phenyl-1H-pyrazolo[3,4-*d*]pyrimidin-4-yl)imino)pentan-2-one (**6d**)

White solid; yield 69%; m.p. 240–242 °C; IR (KBr, ν, cm-1): 3065 (CH- aromatic), 2982 (CH-aliphatic), 1655 (C=N), 1560 (C=C). ^1^H-NMR (DMSO-*d*_6_): δ ppm: 8.98 (d, 2H, J = 7.90 Hz, phenyl-H2, H6), 8.49 (s,1H, -pyrazol), 7.58-7.54 (m, 3H, phenyl-H3, H4, H5), 6.10 (m, 3H), 4.15 (s, 2H, NH-CH2-), 2.51 (s, 3H, pyrimedine-CH3), 2.43 (s, 3H-pyrazol-(CH3)), ^13^C-NMR (DMSO,): 156.59, 154.90, 152.82,150.66,140.96,138.50, 135.57,133.09, 129.79, 128.13, 127.41, 126.11, 124.07, 109.78, 13.24. MS (*m*/*z*): 307.15 (18.4%,M+); Anal. Calcd. for C_17_H_17_N_5_O (307.15): C, 67.09; H, 5.30; N, 27.61%. Found: C, 67.12; H, 5.28; N, 26.63%.

##### General Procedure for the Synthesis of Pyrrole-2,5-Dione Derivatives (**7**,**8**)

To a solution of hydrazinyl derivative (**3**, 0.01 mol) in glacial acetic acid (20 mL), dihydrofuran-2,5-dione, furan-2,5-dione, and indoline-2,3-dione (0.01 mol) were added. The reaction was refluxed at optimum temperature for 16 h. After completion of the reaction mixture (as indicated by TLC), the mixture was concentrated in vacuo and allowed to cool down. The formed solid was filtered and crystallized from ethanol to produce target compounds **7**–**9**. The physical and spectral data of compounds **7**–**9** were as follows:1-((6-Methyl-1-phenyl-1H-pyrazolo[3,4-*d*]pyrimidin-4-yl)amino)pyrrolidine-2,5-dione (**7**)

Blue solid; yield 72%; m.p. 280–282 °C; IR (KBr) *υ* (cm^−1^): 3450 (NH), 3073, 2965 (CH), 1662 (2 C=O); ^1^H-NMR (DMSO-*d*_6_): δ ppm: 9.80 (s, 1H, NH, exchanged with D_2_O), 8.26 (d, 2H, *J* = 7.90 Hz, phenyl-H2, H6), 8.18 (s, 1H, -pyrazol), 7.58–7.54 (m, 3H, phenyl-H3, H4, H5), 2.64 (t, 4H, *J* = 7.90 Hz, pyrolodin-H3, H4), 2.51 (s, 3H, pyrimedine-CH3); ^13^C NMR (DMSO,): 171.00, 168.4, 164.5, 148.00, 139.7, 134.4, 129.3, 126.2, 119.9, 103.03, 30.00, 24.5; MS (*m*/*z*): 322.33 (39.48%,M**^+^**); Anal. Calcd. For C_16_H_14_N_6_O_2_ (322.33): C, 59.62; H, 4.38; N, 26.07%. Found: C, 59.89; H, 4.40; N, 26.12%.

1-((6-Methyl-1-phenyl-1H-pyrazolo[3,4-*d*]pyrimidin-4-yl)amino)-1H-pyrrole-2,5-dione (**8**)

White solid; yield 76%; m.p. 289–290 °C; IR (KBr) *υ* (cm^−1^): 3449 (NH), 3073, 2945 (CH), 1662 (2 C=O); ^1^H NMR (DMSO-*d*_6_): δ ppm: 9.80 (s, 1H, NH, exchanged with D_2_O), 8.26 (d, 2H, *J* = 7.90 Hz, phenyl-H2, H6), 8.18 (s, 1H, -pyrazol), 7.82 (d, 2H, *J* = 7.92 Hz, pyrole-H3, H4), 7.56–7.58 (m, 3H, phenyl-H3, H4, H5), 2.50 (s, 3H, pyrimedine-CH3). ^13^C-NMR (DMSO): 171.00, 168.4, 164.5, 148.00, 139.7, 134.4, 129.3, 126.2, 119.9, 103.03, 30.00, 24.5. MS (*m*/*z*): 320.31 (31.96%,M**^+^**); Anal. Calcd. for C_16_H_12_N_6_O_2_ (320.35): C, 60.00; H, 3.78; N, 26.24%. Found: C, 60.14; H, 3.76; N, 26.28%.

General Procedure for the Synthesis of N-alkyl/aryl-2-(6-methyl-1-phenyl-1H-pyrazolo[3,4-*d*]pyrimidin-4-yl)hydrazinecarbothioamides (**9a**–**e**)

To a solution of hydrazinyl derivative (**3**) (0.01 mol) in butanol (20 mL), appropriate isothiocyanates (0.01 mol), namely, ethyl isothiocyanate, propyl isothiocyanate, butyl isocyanate, vinyl isothiocyanate, and phenyl isothiocyanate (0.01 mol), were added drop wise at 0 °C. The mixture was stirred for 7–10 h at room temperature. Then, the solvent was evaporated under vacuo. The crude product was crystallized from ethanol to afford the corresponding compounds **9a**–**e**, respectively. The physical and spectral data of compounds **9a**–**e** were as follows:N-Ethyl-2-(6-methyl-1-phenyl-1H-pyrazolo[3,4-*d*]pyrimidin-4-yl)hydrazinecarbothioamide (**9a**)

Yellowish-white solid; yield 72%; m.p. 278–280 °C; IR (KBr) *υ* (cm^−1^): 3292 (NH), 3079, 2972 (CH), 1520 (C=C), 1655 (N=C); ^1^H-NMR (DMSO-*d*_6_): δ ppm: 10.09 (s, 1H, NH-NH-C=S), 9.15 (s, 1H, NH-NH-C=S), 8.26 (d, 2H, *J* = 7.90 Hz, phenyl-H2, H6), 8.19 (s, 1H, -pyrazol), 7.59–7.58 (m, 3H, phenyl-H3, H4, H5), 7.30 (s, 1H, C=S-NH-CH2), 4.16 (q, 2H, *J* = 7.2 Hz, -CH2CH3), 2.38 (s, 3H, pyrimidine-CH3), 1.20 (t, 3H, *J* = 7.2, -CH2CH3); ^13^C NMR (DMSO,): 182.52, 168.4, 164.5, 148.00, 139.16, 134.50, 129.66, 126.78, 121.37,103.3, 40.45, 26.25, 14.84. MS (*m*/*z*): 327.41 (13.99%, M**^+^**), 89.33 (100%); Anal. Calcd. for C_15_H_17_N_7_S (327.41): C, 55.03; H, 5.23; N, 29.59%. Found: C, 55.24; H, 5.21; N, 29.83%.

2-(6-Methyl-1-phenyl-1H-pyrazolo[3,4-*d*]pyrimidin-4-yl)-N-propylhydrazinecarbothioamide (**9b**)

White solid; yield 74%; m.p. 280–282 °C; IR (KBr) *υ* (cm^−1^): 3260 (NH), 3067, 2904 (CH), 1527 (C=C), 1658 (N=C); ^1^H-NMR (DMSO-*d*_6_): δ ppm; 9.98 (s, 1H, NH-NH-C=S), 9.84 (s, 1H, NH-NH-C=S), 8.21 (d, 2H, *J* = 7.90 Hz, phenyl-H2, H6), 8.19 (s,1H, -pyrazol), 7.58-7.54 (m, 3H, phenyl-H3, H4, H5), 7.37 (s, 1H, C=S-NH-CH2), 3.33 (t, 2H, *J* = 7.2 Hz, NH-CH2-), 2.58 (m, 2H, *J* = 7.2 Hz, NH-CH2-CH3), 2.38 (s, 3H, pyrimidine-CH3), 1.04 (t, 3H, *J* = 6.9, -CH2CH3); ^13^C-NMR (DMSO,): 184.20, 168.40, 164.50, 148.00, 139.70, 134.30, 129.30, 126.20, 119.9, 103.30, 46.40, 24.50, 23.30, 11.80; MS (*m*/*z*): 341.44 (15.14%, M**^+^**); Anal. Calcd. For C_16_H_19_N_7_S (341.44): C, 56.28; H, 5.61; N, 28.72%. Found: C, 56.50; H, 5.63; N, 28.75%.

N-Butyl-2-(6-methyl-1-phenyl-1H-pyrazolo[3,4-*d*]pyrimidin-4-yl)hydrazinecarbothioamide (**9c**)

Grayish crystals; yield 75%; m.p. 270–272 °C; IR (KBr) *υ* (cm^−1^): 3320 (NH), 3080, 2970 (CH), 1527 (C=C), 1658 (N=C); ^1^H-NMR (DMSO-*d*_6_): δ ppm: 10.09 (s, 1H, NH-NH-C=S), 9.25 (s, 1H, NH-NH-C=S), 8.36 (d, 2H, *J* = 7.90 Hz, phenyl-H2, H6), 8.28 (s, 1H, -pyrazol), 7.60-7.58 (m, 3H, phenyl-H3, H4, H5), 7.32 (s, 1H, C=S-NH-CH2), 4.13 (t, 2H, *J* = 7.2 Hz, NH-CH2CH2), 2.51 (s, 3H, pyrimidine-CH3), 1.30 (m, 2H, NH-CH2CH2),1.17 (q, 2H, CH2-CH2-CH2CH3), 0.89 (t, 3H, *J* = 7.2 Hz, -CH2CH2-CH2CH3); ^13^C-NMR (DMSO,)): 184.20, 168.40, 164.50, 148.00, 139.70, 134.30, 129.30, 126.20, 119.9, 103.30, 46.40, 32.40, 24.50, 20.30, 20,40; MS (*m*/*z*): 341.44 (15.14%, M**^+^**), Anal. Calcd. for C_17_H_21_N_7_S (355.46): C, 57.44; H, 5.96; N, 27.58%. Found: C, 57.70; H, 5.98; N, 27.60%.

2-(6-Methyl-1-phenyl-1H-pyrazolo[3,4-*d*]pyrimidin-4-yl)-N-vinylhydrazine-1-carbothioamide (**9d**)

White solid; yield 70%; m.p. 268–270 °C; IR (KBr) *υ* (cm^−1^): 3356 (NH), 3081, 2908 (CH), 1570 (C=C), 1620 (C=N); ^1^H-NMR (DMSO-*d*_6_): δ ppm: 10.20 (s, 1H, NH-NH-C=S-NH), 8.30 (d, 2H, *J* = 7.90 Hz, phenyl-H2, H6), 8.17 (s, 1H, -pyrazol), 8.10 (d, 1H, *J* = 7.90 Hz, NH-NH-C=S), 7.82 (t, 1H, *J* = 7.90 Hz, S=C-NH-CH2-), 7.58–7.56 (m, 3H, phenyl-H3, H4, H5), 6.80 (m, 1H, *J* = 7.30, NH-CH2-CH=CH2), 6.10 (t, 1H, *J* = 7.2, C=S-NH-CH2=CH), 5.85 (d, 1H, *J* = 7.2 Hz, C=S-NHCH2-CH=CH), 2.48 (s, 3H, pyrimidine-CH3). ^13^C-NMR (DMSO): 184.22, 168.42, 164.52, 148.00, 141.00, 139.76, 134.32, 129.30, 126.22, 119, 93.3, 46.40, 24.50, 20.30.

2-(6-Methyl-1-phenyl-1H-pyrazolo[3,4-*d*]pyrimidin-4-yl)-N-phenylhydrazinecarbothioamide (**9e**)

Brown solid; yield 66%; m.p. 281–282 °C; IR (KBr) *υ* (cm^−1^): 3265 (NH), 3026 (aromatic-CH), 2924 (aliphatic-CH), 1510 (C=C), 1530 (C=N); ^1^H-NMR (DMSO-*d*_6_): δ ppm: 10.13 (s, 1H, S=C-NH), 9.81 (s, 1H, -NH-S=C-NH), 9.13 (s, 1H, NH-NH-S=C), 8.98 (d, 2H, *J* = 7.90 Hz, phenyl-H2, H6), 8.49 (s, 1H, -pyrazol), 7.70 (d, 2H, *J* = 7.90 Hz,-NH-phenyl-H2, H6), 7.58–7.54 (m, 3H, phenyl-H3, H4, H5), 7.49-7.36 (m, 3H, NH-phenyl-H3, H4, H5), 2.51 (s, 3H, pyrimedine-CH3). ^13^C-NMR (DMSO): 161.22, 156.03, 139.95, 139.30, 134.98, 129.61, 129.68, 129.12, 126.67, 122.67, 121.40, 98.97,40.55, 40.35, 39.93, 39.51, 26.33. MS (*m*/*z*): 375.45 (M^+^, 13.70%), Anal. Calcd.for C_19_H_17_N_7_S (375.45): C, 60.78; H, 4.56; N, 26.11%; Found: C, 61.05; H, 4.73;N, 26.16%, purity by HPLC = 98.84%.

### 4.2. Biological Evaluation

#### 4.2.1. Cytotoxicity Evaluation Using Viability Assay

The cytotoxic activity was assessed using the 3-(4,5-dimethylthiazol-2-yl)-2,5-diphenyl tetrazolium bromide (MTT) colorimetric assay, as reported previously [62,63]. Doxorubicin was used as a reference standard, and DMSO was used as a negative control. Briefly, the designated cell lines were cultured at a concentration of 5 × 10^4^ cell/well in 96-well plates. The cells were incubated for 24 h under optimum conditions. Following this, the designated molecules were introduced (in serial dilutions from 0 to 50 μg/mL), and the cells were incubated for an additional 24 h. Following this, the media were withdrawn, and 100 μL of fresh medium was added. Then, 10 µL of the 12 mM MTT stock solution was administered to each well. The plates were then incubated for 4 h. Then, the cells were exposed to different compounds at the desired concentrations (0.01, 0.1, 1, 10, and 100 µM) or to 1% dimethyl sulfoxide (DMSO) for 48 and 72 h. An 85-μL aliquot of the media was removed from the wells, and 50 µL of DMSO was added to each well and mixed thoroughly with the pipette and incubated at 37 °C for 10 min. Then, the optical density was measured at 590 nm with the microplate reader (Sunrise, TECAN, Inc., Morrisville, NC, USA).

#### 4.2.2. In Vitro Enzyme Inhibitory Assays (against EGFR^wt^, EGFR^T790M^, VEGFR-2, and Topo II)

Enzyme inhibitory assays for the designated compounds **5b**, 5i, and 9e were carried out in triplicate, as described earlier [64]. The assays were conducted in Vacsera (Giza, Egypt). Curve fitting was performed using GraphPad Prism. Data are represented as means ± SD from three independent experiments.

#### 4.2.3. Quantification of Apoptosis by Flow Cytometry

Annexin V-FITC apoptosis assay was conducted utilizing the Annexin V-FITC/PI double staining detection kit (BD Pharmingen, San Diego, CA, USA), as described elsewhere [65,66].

#### 4.2.4. Cell Cycle Analysis

Cell cycle arrest and distribution was evaluated using the Propidium Iodide Flow Cytometry Kit followed by flow cytometry analysis, as described previously [67].

### 4.3. Molecular Docking

The crystal structures of EGFR (PDB ID: 1M17) and VGFR-2 (PDB ID: 3EWH) were extracted from Protein Data Bank (http://www.pdb.org (accessed on 25 September 2023)). The AutoDock 3.0 [68] and the MOE software 2008 [69] were used to perform all calculations on compound **5i**. The protocol in detailed [34] was adopted.

## Data Availability

All data are provided in the article or Appendix A.

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
