# Peer review of "Development of Novel Class of Phenylpyrazolo[3,4-*d*]pyrimidine-Based Analogs with Potent Anticancer Activity and Multitarget Enzyme Inhibition Supported by Docking Studies"

_ijms, 2023, doi:10.3390/ijms241915026_

Round 1
Reviewer 1 Report (Previous Reviewer 1)
Manuscript ID: ijms-2647906
The Article " Development of Novel Class of phenylpyrazolo[3,4-d]pyrimidine based Analogs with Potent Anticancer Activity and Multitarget Enzyme Inhibition Supported by Docking Studies" by Hany E. A. Ahmed et. al
This revised manuscript represents a resubmission by the authors, meticulously enhanced in response to the invaluable feedback provided by the esteemed reviewers. The meticulous revisions and refinements made in this iteration unequivocally affirm its readiness for publication in this esteemed journal. The authors have diligently addressed the reviewers' comments, significantly improving the overall quality and rigor of the manuscript.
Author Response
Thanks for the respected comment.
Reviewer 2 Report (Previous Reviewer 2)
In this revised manuscript, the authors addressed most of the minor issues in the previous version. Please consider the following points for an improved version:
1. Please provide IC50 of 5b/5i/93 on healthy cell line in Table 2. And also include the dose-dependent curves in Figure 4.
2. In this manuscript, μg/mL was used here and there as the unit of compound concentration for IC50 values. The authors corrected the units in the main text but not in the abstract. And also, IC50 unit is misused in the conclusion part.
Author Response
- Please provide IC50 of 5b/5i/93 on healthy cell line in Table 2. And also include the dose-dependent curves in Figure 4.
- We thank you very much. We added the normal cytotoxicity of the most potent compounds 5b, 5i, and 9e analogs to Table 2. In addition, all dose-response curves are inserted in the supporting material and kept ONLY curves of active analogs within MS to reduce the crowdedness within MS.
- In this manuscript, μg/mL was used here and there as the unit of compound concentration for IC50 values. The authors corrected the units in the main text but not in the abstract. And also, IC50 unit is misused in the conclusion part.
- Thanks for the respected comment. We ran all the MS and corrected to uM.
Reviewer 3 Report (Previous Reviewer 3)
The authors replied to most of my previous comments, however most of the HRMS results showed impure compounds which is unaccepted.
They should provide the integration of the 1H NMR peaks which are missing for most of the compounds and the HRMS results should show only one peak (the product's peak only).
N.B. I clarified this requirement in my last review report.
No comments
Author Response
- They should provide the integration of the 1H NMR peaks which are missing for most of the compounds and the HRMS results should show only one peak (the product's peak only).
- Thanks for the respected comment. We made all peak integration for all compounds as appeared in supporting and MS. Regarding compound purity, we made HNMR, CNMR, CHN, and MS to confirm the structure with more than crystallization experiments. In addition, we made HPLC analysis for the most active compounds as more experiments for confirming compound purity with more than 95% purity. The charts are included in the supporting materials and results are added to the manuscript. We have no funds to do extra HRMS for the compounds and at least in this paper, the spectroscopic data are enough.
Round 2
Reviewer 2 Report (Previous Reviewer 2)
1. IC50 values and curves hadn’t been added into Table2 and so for data point curves in Fig 4 in the revised version. For normal cell lines, it is acceptable to show as >35uM.
2. Why the highest concentration tested on normal cell line changed from 80uM to 35uM?
Author Response
- IC50 values and curves hadn’t been added into Table2 and so for data point curves in Fig 4 in the revised version. For normal cell lines, it is acceptable to show as >35uM.
- We thank you, we added all curves within the MS in Figure 4. Moreover, for normal cell line cytotoxicity, we added three active analogs to table and mentioned in text>35 uM.
- Why the highest concentration tested on normal cell line changed from 80uM to 35uM?
- Thanks for you, we wrote it by mistake, but corrected in MS to be 35 uM.
Reviewer 3 Report (Previous Reviewer 3)
The authors have replied to my previous comments, no further comments to add.
No comments
Author Response
Thanks for you and all support by your valuable comments in improving our MS.
Round 3
Reviewer 2 Report (Previous Reviewer 2)
Issues have been addressed.
This manuscript is a resubmission of an earlier submission. The following is a list of the peer review reports and author responses from that submission.
Round 1
Reviewer 1 Report
Manuscript ID:ijms-2592813
The Article " Development of Novel Class of phenylpyrazolo[3,4-d]pyrimidine based Analogs with Potent Anticancer Activity and Multitarget Enzyme Inhibition Supported by Docking Studies" by Hany E. A. Ahmed et. al
Authors have disclosed their recent results on phenylpyrazolo[3,4-d]pyrimidine derivatives and its biological studies, however authors should be careful about providing supporting data of synthesized compounds and its supporting information's provided in the manuscript.
This paper is not recommended for this journal based on the information provided by the authors.
Reviewer 3 Report
This paper aims to investigate the development of new anticancer agents and multi-target enzyme inhibitors by the design and synthesis of phenylpyrazolo[3,4-d]pyrimidine derivatives.
The manuscript is written comprehensively enough to be understandable; the authors addressed this aim synthesizing four different series of hydrazide, methylhydrazide, closed ring systems, and thiourea derivatives of high accessibility and good yields which are then evaluated for their bioactivity. Authors also provided the mechanistic analysis using EGFR, VGFR2, and Top-II enzymatic assay and cell cycle and apoptotic analyses.
The paper stated the purpose, discussion and global implication are clearly stated and consistent with the rest of the manuscript; authors provided enough information in their discussion by using a good number of important articles talked about the subject as well as using clear figures to support their discussion and they used the molecular docking to explore the binding mode and mechanism of their compounds on protein targets. However, they did not prove the purity of their synthetic compounds by providing the integration of each peak in the 1H NMR spectrums and the data is missing the HMRS data/figures in the supplementary info, these two parameters are essential to prove the compounds’ purity.
The authors addressed their hypothesis and opinion in a reproducible way which facilitate in reaching a conclusion elucidates that three of their synthetic compounds were potent analogues with IC50 values of 3-10 µg/mL, range cytotoxicity. Moreover, most of the compounds had stronger anti-proliferative activity proved by the provided mechanistic analysis on the cancer cell line when compared to the positive control medication. One of their compounds 5i enhanced apoptosis in MCF-7 cancer cells and appears to be promising as a novel multi pyrazolopyrimidine-based lead compound for the identification of new anticancer medicines that target the EGFR/VGFR2 enzymes.
The abbreviations should be explained at the first place they are mentioned.
No plagiarism has been detected.
References: The authors should follow the journal guidelines for their references.
Ref 4, 26, 32 and some other references missed the journal name.
No comments
Reviewer 4 Report
In this manuscript the authors have reported the preparation of a series of phenylpyrazolo[3,4-d]pyrimidine evaluated for their anticancer effects on a panel of cancer cell lines. Some of these compounds showed potent cytotoxicity with variable degrees of potency and cell lines selectivity in anti-proliferative assays with low resistance. The data obtained revealed that most of these compounds were dual EGFR/VGFR2 inhibitors at IC50 value range; 0.13-6.09 μg/mL. Among these, compound 5i was the most potent non-selective dual EGFR/VGFR2 inhibitor with inhibitory concentrations of 0.29 and 2.60 μg/mL respectively.
Although the manuscript falls within the aim and scope of this journal, the research results reported are premature for publication. There are some errors exist in this version of manuscript, and to make this paper more clear, this manuscript needs to be revised carefully. The manuscript suffers from important deficiencies in terms of clarity of the presentation of the results, the rationale behind the design of these compounds. The exprerimental section is particularly chaotic. For several compounds, such as derivative 3, two different 1H and 13C NMR data were reported. So, in my opinion, the submission does not meet the standards of this journal.
I include some other issues that maybe useful for the authors when submitting elsewhere or for a resubmission.
All compounds of Figure 1 were characterized by the presence of pyrimidine alone or in the byciclic system such as quinazoline. The sentences at lines 92-96 should be revised.
An EGFR inhibitor (such as Erlotinib or Geftinib) should be employed as reference compound.
TThe inhibitory activity of the selected compounds should be evaluated on against EGFRWT and EGFRT790M
dSimilar compounds have been previously published (J Enzyme Inhib Med Chem. 2022; 37(1): 2283–2303.)
eThe in vitro inhibitory activities against EGFR and VEGFR should be reported at micromolar concentrations.
f1H and 13C NMR data should be reported for all final synthesized compounds.